# Costs of Point-of-Care Viral Load Testing for Adults and Children Living with HIV in Kenya

**DOI:** 10.3390/diagnostics11010140

**Published:** 2021-01-19

**Authors:** Michelle Ann Bulterys, Patrick Oyaro, Evelyn Brown, Nashon Yongo, Enericah Karauki, James Wagude, Leonard Kingwara, Nancy Bowen, Susan Njogo, Anjuli D. Wagner, Irene Mukui, Frederick Oluoch, Lisa Abuogi, Rena Patel, Monisha Sharma

**Affiliations:** 1Departments of Epidemiology and Global Health, University of Washington, Seattle, WA 98105, USA; patrickoyaro@gmail.com (P.O.); anjuliw@uw.edu (A.D.W.); imukui@uw.edu (I.M.); rcpatel@uw.edu (R.P.); msharma1@uw.edu (M.S.); 2Health Innovations Kenya, Kisumu, Kenya; 3Department of HIV Research, University of Washington Kenya, Nairobi, Kenya; evelynbrown@uwkenya.org (E.B.); nashyong@yahoo.com (N.Y.); enericahkarauki@gmail.com (E.K.); 4Department of Health, Siaya County, Kenya; jwagude@gmail.com; 5National HIV Reference Laboratory, National Public Health Laboratory, Nairobi, Kenya; leonard.kingwara@gmail.com (L.K.); njebungeibowen@gmail.com (N.B.); fredkoluoch@gmail.com (F.O.); 6National AIDS and STI Control Programme, Ministry of Health, Nairobi 19361, Kenya; smnjogo2012@gmail.com; 7Department of Pediatrics, University of Colorado, Denver, CO 80045, USA; lisa.abuogi@childrenscolorado.org; 8Department of Medicine, Division of Allergy and Infectious Diseases, University of Washington, Seattle, WA 98105, USA

**Keywords:** costing, point-of-care viral load testing, viral load test, HIV care, HIV management

## Abstract

Background: The number of people living with HIV (PLHIV) in need of treatment monitoring in low-and-middle-income countries is rapidly expanding, straining existing laboratory capacity. Point-of-care viral load (POC VL) testing can alleviate the burden on centralized laboratories and enable faster delivery of results, improving clinical outcomes. However, implementation costs are uncertain and will depend on clinic testing volume. We sought to estimate the costs of decentralized POC VL testing compared to centralized laboratory testing for adults and children receiving HIV care in Kenya. Methods: We conducted microcosting to estimate the per-patient costs of POC VL testing compared to known costs of centralized laboratory testing. We completed time-and-motion observations and stakeholder interviews to assess personnel structures, staff time, equipment costs, and laboratory processes associated with POC VL administration. Capital costs were estimated using a 5 year lifespan and a 3% annual discount rate. Results: We estimated that POC VL testing cost USD $24.25 per test, assuming a clinic is conducting 100 VL tests per month. Test cartridge and laboratory equipment costs accounted for most of the cost (62% and 28%, respectively). Costs varied by number of VL tests conducted at the clinic, ranging from $54.93 to $18.12 per test assuming 20 to 500 VL tests per month, respectively. A VL test processed at a centralized laboratory was estimated to cost USD $25.65. Conclusion: POC VL testing for HIV treatment monitoring can be feasibly implemented in clinics within Kenya and costs declined with higher testing volumes. Our cost estimates are useful to policymakers in planning resource allocation and can inform cost-effectiveness analyses evaluating POC VL testing.

## 1. Introduction

In 2019, an estimated 38 million people were living with HIV (PLHIV) worldwide, with the vast majority residing in resource-limited settings [1]. The scale-up of universal HIV treatment has rapidly increased the number of PLHIV on antiretroviral therapy (ART). In eastern and southern Africa, 67% of adults living with HIV and 58% of children living with HIV (CLHIV) aged 0–14 years were accessing ART [1]. Monitoring the millions of PLHIV on ART is challenging in resource-limited settings. The World Health Organization (WHO) recommends routine testing of HIV viral load (VL) to evaluate to monitor treatment response, adherence, and virologic suppression among patients receiving ART [2]. However, fewer than 50% of PLHIV on ART in sub-Saharan Africa receive routine VL monitoring [3]. The vast majority of VL tests in sub-Saharan Africa are processed in centralized laboratories that require highly trained staff and specialized equipment. Scale-up is hindered by challenges of timely sample transportation and inadequate infrastructure [4]. In addition, for patients who receive VL testing, results are often returned to the patient at a subsequent clinic visit after several weeks, leading to delays in clinical decision making, non-delivery of results, or lack of provision of adherence counseling [3]. Expanding the availability of routine VL testing is essential for delivering high-quality ART care, improving clinical outcomes, ensuring the longevity of existing ART regimens, and preventing HIV transmission [5].

Point-of-care (POC) VL testing is a promising strategy recommended by the WHO that can increase VL testing coverage and provide rapid delivery of results [6,7]. POC can enable faster identification of individuals with treatment failure for targeted adherence counseling and/or regimen switching. POC testing has been shown in randomized trials to improve viral suppression and retention in care compared to centralized laboratory testing [7], and is projected to be cost-effective in other sub-Saharan African settings [8,9].

Kenya is one of the countries hardest hit by the HIV epidemic, with an adult HIV prevalence of 4.9% [10]. There are approximately 1.6 million PLHIV in Kenya, and Kisumu County in western Kenya contributes substantially to the national HIV burden with an adult prevalence of 17.5% [10]. Kenya was one of the first countries in Africa to scale up routine viral load testing programs, and is currently using a high-volume centralized testing model, in which patients receive their results at their next scheduled routine visit (median 21 days after sample collection) [11,12]. In Kenya, children with HIV are more likely to have elevated viral loads than adults [12]. Implementing nation-wide POC VL testing platforms to complement existing centralized laboratory testing can alleviate the burden on centralized laboratories and reduce delays by enabling the rapid return of VL results and counseling within the same clinical visit [13]. The cost of implementing POC VL testing in Kenya is uncertain. We aimed to estimate the cost of implementing decentralized POC testing for VL monitoring among children and adults in Kenya, compared to routine VL testing at centralized public laboratories.

## 2. Materials and Methods

### 2.1. The Opt4Kids and Opt4Mamas Studies

The present costing study was conducted alongside the ongoing randomized controlled trial (Opt4Kids, ClinicalTrials.gov Identifier: NCT03820323) and the parallel cohort study (Opt4Mamas), which are evaluating the impact of POC VL testing for children aged 1–14 years and pregnant women, respectively. The Opt4Kids study enrolled participants from March to December 2019, and the Opt4Mamas study from February 2019 to June 2020. Study participants are receiving HIV care at five Ministry of Health facilities in western Kenya. Study procedures for Opt4Kids have been previously described and are similar in Opt4Mamas [14]. Briefly, children enrolling at each site from March to December 2019 were randomized to intervention (POC VL testing) vs. standard of care (SOC VL testing) arms and followed for 12 months. In the Opt4Mamas study, 350 pregnant women were enrolled in SOC VL testing and 350 pregnant women were enrolled in POC VL testing and were followed through 6 months postpartum. The primary outcome of the Opt4Kids study is the proportion of children achieving VL suppression 12 months after POC VL testing implementation. Outcomes in the Opt4Mamas study include the proportion of women achieving VL suppression at 6 months postpartum.

### 2.2. POC VL Instrument

POC VL testing in the studies is conducted via the Xpert^®^ HIV-1 VL Assay developed by Cepheid, which runs on GeneXpert^®^ systems (Cepheid, Sunnyvale, USA). It has been validated and found feasible and reliable in rural African communities [15]. This assay uses reverse transcriptase polymerase chain reaction (PCR) technology to detect HIV in 1 mL of plasma over a range of 40 to 10 million copies/mL, with 94% sensitivity and 99% specificity, has an average run time of 90 min, and is a self-contained kit that includes the necessary PCR reagents and supplies [16]. Different sized Xpert^®^ equipment, herein referred to as “POC equipment” can be purchased to accommodate different numbers of simultaneous tests (ranging from 4 to 80 samples processed at the same time), greatly impacting the cost of the machine. For this costing analysis, we assumed POC equipment would run 4 samples simultaneously. The 4-module machine was used by three of the four on-site laboratories participating in this study and is the most commonly utilized GeneXpert^®^ equipment in Kenyan clinics. Further, the 4-module machine was identified as the most cost-effective size for the average clinic testing volume based on calculations by the Kenyan Ministry of Health National Laboratory at the time that national roll-out began (personal communications with Dr. Leonard Kingwara, Head of National HIV Reference Laboratory).

### 2.3. Micro-Costing Study Design

We conducted detailed microcosting from the provider perspective to compare the per-test cost of administering an HIV POC VL test to a centralized VL test from the payer perspective (Kenyan Ministry of Health (MOH)). Costs (2020 USD) were collected from expense reports, staff and expert interviews, literature, and were divided into personnel, clinical supplies, equipment, office supplies, and laboratory space. All blood samples for POC VL and centralized laboratory tests were obtained through venous blood draws. We assumed that the following would remain the same across POC and centralized VL testing: personnel time to draw blood from client, supplies needed to collect blood samples, and office supplies. We assumed clinics would purchase a 4-module processing GeneXpert IV machine, which would be housed in the clinic laboratory. We assumed a 5-year instrument life and a 3% annual discount rate for POC equipment costs. Costs were collected in Kenyan Shillings and converted to USD using the exchange rate of 107.77 Kenyan Shillings to USD $1 reported at the midpoint of the data collection time period (August 2020). We did not include societal costs such as those incurred by patients to attend a clinic visit, including transport and opportunity costs of missed work.

### 2.4. POC VL Testing Costs and Assumptions

We developed an initial list of main activities by reviewing the parent study protocols [14] and discussing with the site team. We conducted semi-structured interviews with the study, facility, national laboratory, and Kenyan MOH staff to obtain information on resource use and staff time needed to deliver POC VL testing in public facilities. Unit costs for administering blood draw and viral load testing were estimated separately for children and adults, and unit costs for post-test counseling were estimated separately for clients on ART who were virally suppressed and unsuppressed. Based on our interviews, we assumed blood samples were collected and processed by a laboratory technician and post-test counseling was done by an adherence counselor or HIV care provider in the HIV clinic, for both the POC and centralized laboratory testing platforms. Based on estimates published by the Clinton Health Access Initiative (CHAI), we estimated that an individual POC cartridge costs USD $14.90 according to in-country negotiations [17]. We assumed that implementing POC VL testing would require a three-day training for staff at the facility, and would involve a GeneXpert County Representative (trainer), laboratory supervisor, laboratory technician, a nurse, and adherence counselor. Personnel costs for this training are included in the analysis.

We assumed that POC test material and supplies reflect preferential pricing provided by manufactures for low and middle-income countries (LMICs). As the equipment component of the cost per test depends on the number of tests conducted, we calculated POC VL test cost per patient at varying VL testing volumes. The lower bound of the testing volume range was estimated via communications with clinic staff, and the upper bound was estimated as the maximum capacity of POC VL tests that the machine could run in a month, assuming that machines were only used for POC VL testing. Assuming one 4-cartridge GeneXpert machine, the maximum capacity would be 24 tests per 7-h workday, 480 per month, and 5760 tests per year.

Based on conversations with in-country experts, we assumed clinics would not have a stable power supply (works approximately 60% of the time), and therefore included costs of a small back-up generator and battery for the GeneXpert equipment. Although most back-up generators can serve multiple machines (including larger POC machines) and refrigerators in a laboratory, hence sharing the costs across these units, we included the full cost as part of the start-up. Without a back-up generator, we estimate that sample wastage occurs 15% of the time compared to 1% of the time with back-up generator. We estimated electricity costs using published data on the average energy consumption by a GeneXpert instrument in low and middle-income countries [18]. The average annual energy consumption for a 4-module GeneXpert instrument is estimated to be 489 Kilowatts (KWh) if kept on during the entire workday [18], which would cost approximately $106.93 per year in Kenya [19]. Assuming the 4-module machine could run up to 5760 tests per year, we estimate each test would use $0.02 of energy. We estimated that the 4-module instrument takes up 9 square feet, and the monthly cost to rent space in Kenya was estimated to be $0.44 per square foot (total monthly rent for space = $3.96); the per-test cost of space was assumed to be dependent on monthly testing volume [20].

POC testing platforms already exist at several facilities across Kenya and are primarily used for diagnostic testing for tuberculosis (TB) and other diseases. The national prevalence of TB in Kenya is estimated to be approximately 558 per 100,000 adults [21], however, TB prevalence varies substantially by region. Therefore, we conducted a sensitivity analysis to assess the potential impact of cost-sharing of POC GeneXpert instruments with TB diagnostics, so that regions can adjust these estimates by their overall testing volume. Cost-sharing with other disease diagnostics using the POC platform is possible (e.g., STIs, malaria), but an infrastructure outside of TB does not currently robustly exist. Therefore, we assume cost-sharing would be done by leveraging existing TB infrastructures. As POC VL is not yet fully rolled out in Kenya, the potential for cost-sharing equipment with other diseases remains uncertain. Therefore, we present costs assuming 25%, 50%, and 75% of equipment cost-sharing with TB diagnostics programs. This scenario assumes that a higher proportion of the equipment sharing lowers the cost of VL testing, shared costs are absorbed by TB programs.

### 2.5. Centralized Laboratory Costs and Assumptions

The fee-for-service cost for conducting a centralized laboratory VL test in the Kenya public health sector (USD $24.63) was obtained from USAID data; [11] this cost includes sample transport, laboratory staffing costs, laboratory consumables and equipment, instruments, and maintenance. Centralized laboratory costs were calculated using the previously estimated fee-for-service cost from a USAID analysis that was charged to the clinic per test, plus the costs of personnel time and blood draw supplies required to collect a sample, which was obtained from our microcosting and time and motion observations.

When viral load tests repeatedly indicate poor management of HIV treatment, and poor adherence is ruled out, regardless of the viral load testing method (POC vs. centralized testing), a drug resistance test (DRT) can be requested. The per-test cost to run a DRT at the National HIV Reference Laboratory in Nairobi, Kenya, is USD $67.64 [22]. This fee-for-service cost includes personnel time to run all laboratory processes, cost of laboratory equipment and supplies, and protection (e.g., gloves). This cost does not account for costs to collect the blood sample and transport the sample to the centralized laboratory (e.g., personnel time for blood sample and transport, clinical supplies for blood collection, ice storage and cooler for transport, and refrigeration for storage at the laboratory).

### 2.6. Time and Motion Observation

We conducted time and motion observations over six weeks in November 2019–August 2020 in study clinics to estimate staff time needed to implement POC VL testing. When a study participant arrived in the clinic room, a stopwatch was started and the amount of time taken for each activity was recorded using pen and paper (e.g., screening, informed consent process, research questionnaires, blood draw, counseling, lab processing for POC VL testing, and post-test adherence counseling upon return of results). Time and motion observations were conducted until we reached information saturation, and data were extracted into Microsoft Excel (Microsoft Corporate Headquarters, Redmond, WA, USA) to calculate total and unit costs associated with the intervention. Saturation of information was defined as occurring when additional observations did not change the estimates regarding how long an activity took. Resources and time spent on research activities (e.g., administering informed consent and study questionnaires) were removed from programmatic costs.

## 3. Results

### Cost per POC VL Test

We estimated that POC VL testing cost USD $24.25 per test, assuming a clinic volume of 100 VL tests conducted per month (Table 1). Costs were substantially higher at lower testing volumes; at clinics conducting 20 VL tests per month POC test costs were estimated to be USD $54.93 per test compared to $18.12 at clinics conducting 500 VL tests per month. The cost of a viral load sample processed in a centralized laboratory was USD $25.65 per test, as calculated by the fee-for-service cost from USAID (USD $24.63 [11]) plus the personnel time and clinical supplies costs needed to collect the blood sample, calculated via our microcosting study and time and motion observations. POC testing costs declined steeply as testing volumes increased from 20 to 100 VL tests per month and then declined more slowly from monthly testing volumes of 100 to 500 (Figure 1). The largest contributors to the per-test cost were the testing cartridge and laboratory equipment, which accounted for 62% and 28% of the total costs, respectively, in a clinic conducting 100 monthly VL tests (Figure 2). A sensitivity analysis varying the cartridge cost demonstrated that per-test POC VL cost drops substantially when the cartridge is cheaper; for example, at a USD $12.00 cost per cartridge, the per-test cost is $21.35 for a clinic volume of 100 monthly VL tests (Table 2). Assuming a cost-sharing scenario in which POC instruments were also utilized for TB diagnostics resulted in lower POC VL test costs. At a clinic conducting 100 VL tests per month, VL test costs were USD $22.53 using 75% instrument use and $19.08 assuming 25% instrument use (Table 1). Cost sharing had a larger impact in reducing VL test costs at lower clinic testing volumes compared to higher testing volumes. For example, 50% cost-sharing reduced VL test costs from USD $54.93 to $37.75 at clinics testing 20 VL per month but only reduced costs from USD $18.12 to $17.42 in clinics conducting 500 VL tests per month. The start-up costs of the GeneXpert equipment are presented in Table 2. The cost of a GeneXpert machine was estimated to be $26,000.00, which annualized to a per-year cost of USD $5677.00. Costs did not differ substantially between adults and children. Time and motion observations on VL testing comparing pregnant women and children and post-test counseling depending on viral suppression status are presented in Table 3.Post-test counseling costs (Table 4) were not included in the main costing analysis (Table 1).

## 4. Discussion

In this microcosting study, we estimated the per-test POC VL costs at varying clinic volumes compared to centralized laboratory testing in Kenya. Overall, we found that the POC VL testing’s per-test cost is higher than centralized laboratory testing, particularly for lower volume clinics. However, POC test costs declined rapidly with higher VL testing volumes since the POC instrument’s cost is spread over a larger number of tests. At clinics conducting 500 VL tests per month or with cartridge costs reducing to $12.00, POC testing had a lower cost than centralized laboratory testing. Overall, we find that POC HIV VL testing can be implemented into routine care at reasonable costs in clinics with moderate to large VL testing loads. Since POC testing costs were highly sensitive to the number of tests performed on each instrument, clinic volumes will be important as countries consider how to scale up POC VL technology to maximize resources within a limited budget. Based on consultation with in-country staff, we assumed a typical large Kenyan clinic conducts 100 viral load tests per month. Our study is among the first to estimate results of counseling costs by viral suppression among children and adults; we find that adherence counseling costs are highest for virally unsuppressed children and lowest for virally suppressed adults.

Our findings are similar to previous costing studies, which estimated the cost per POC VL test to be between $25 and $30 [8,13]. One study in South Africa found that staff time to collect a sample and conduct the POC VL test contributed $8 to the per-test cost, which is substantially higher than our staff costs [23]. Several differences exist between this study and ours: labor costs are higher in South Africa than in Kenya, and staff time was calculated using the total time needed to run a POC test without accounting for waiting time when technicians generally engage in other laboratory tasks [23]. The present study utilized data from detailed time and motion observations to cost only the active time that staff spent involved in POC VL testing, and removed time spent on routine clinical care. A large study that assessed cost trends for HIV treatment services in Botswana, Ethiopia, Nigeria, Uganda, and Vietnam found that per-patient costs were highest at the beginning of service scale-up and dropped rapidly as sites matured (nearly 50% reduction of costs after the first year following scale-up) [24]. Although this large study’s costs accounted for all treatment services including ART, we expect that POC VL testing platforms will also be highest at the time of start-up and will steadily drop as sites reach workflow efficiency over time.

Identifying cost drivers of POC VL testing can inform strategies to reduce costs. We find that the cost of a GeneXpert^®^ cartridge ($14.90/test) makes up the majority of the POC VL costs (72–91%), which is consistent with other costing studies [13,25]. Negotiating reductions in cartridge costs can greatly reduce overall costs for POC VL testing, and may result in POC VL tests being less costly than centralized laboratory testing as we demonstrate in sensitivity analyses. Furthermore, there is potential to integrate HIV POC VL testing into existing GeneXpert^®^ platforms across Kenya that are currently being used to test for TB and other sexually transmitted infections [26]. We find that cost sharing with TB diagnostics reduces the cost of POC VL testing for low volume clinics but has a marginal impact on clinics with high VL testing volumes. Therefore, for rural clinics with smaller patient volumes in which instruments may not be used at capacity, the cost of the GeneXpert^®^ equipment and recurring costs could be shared with other diseases’ diagnostics to improve cost efficiency. Patients attending low-volume clinics in rural settings may benefit from POC testing since poor quality roads can increase transport time to centralized laboratories leading to delayed results, degrading of samples, and increased loss to follow up [13]. Although we utilize an average cost estimate for centralized laboratory VL testing, costs are substantially higher for rural clinics due to the impediment of timely transport of samples to a centralized laboratory. In addition to having a minimum impact on POC VL testing costs, instrument sharing may be more difficult to operationalize in settings with both high TB burden and high HIV prevalence since both diseases would be competing for use of the same machine.

Layering on spoke-and-hub models for low volume or rural facilities may further decrease POC VL costs, which Kenya has already implemented for other disease testing platforms using GeneXpert. A study in Zimbabwe found that among women on Option B+, those who attended antenatal care at smaller volume clinics were 36% less likely to undergo VL testing than women at higher volume hospitals [27]. While larger volume clinics have lower per-test costs per, smaller volume clinics may experience greater health benefits by implementing POC VL testing. A study in Zambia compared costs if POC instruments were placed at all the 10% of facilities that are not part of the centralized laboratory network for VL testing (hardest-to-reach facilities) and found the cost per test was $41.81 [8]. This cost per test dropped to $39.58 (6% reduction) when a proportion of the hardest-to-reach facilities served as POC hubs, increasing the POC instrument utilization [8]. This study concluded that POC VL testing implementation reduced the cost per VL test by 6–35% when combining on-site placement as well as use of POC hubs [8].

The demand for VL testing in 130 predominantly LMICs is projected to increase dramatically from 14.7 million tests in 2017 to over 28.5 million in 2022 [28]. As countries shift focus from CD4 count to VL testing, there is increasing demand for governments and funders to prioritize rapid POC testing to optimize efficiency of patient monitoring. Our cost estimates provide essential data for future economic analyses evaluating the cost-effectiveness and budget impact of POC VL testing for both adults and children living with HIV in Kenya. Although POC VL testing is more costly than centralized laboratory testing, cost savings to the health system may occur by reducing loss to follow up and increasing ART adherence. Indeed, modeling analyses show that POC testing is projected to reduce HIV-related deaths and HIV transmissions and is cost-effective for monitoring treatment in adults living with HIV in South Africa, particularly in facilities with high levels of virologic failure [9,29].

Our study has several limitations. We conducted microcosting for one region of Kenya, which may not generalize nationally to other areas. Furthermore, costs were collected alongside a research trial, which may not be representative of routine clinical practice. However, our time and motion observations included data on routine clinic visits and we excluded research-related time costs. We used published cost estimates for centralized laboratory costs; therefore, we were not able to disaggregate unit costs for comparison to POC VL test costs. Our study did not account for the overhead, administrative and supervisory costs needed to implement POC nationally; future studies are needed to assess POC VL test implementation costs beyond the clinic level. We obtained instrument costs from the Cepheid distributor in Kenya rather than from the Kenyan Ministry of Health, so it is possible that costs estimates may vary depending on the source. However, the costs obtained from the Cepheid distributor reflect the country-negotiated costs and are consistent with the published literature [8,13]. Further, we only estimate the cost of POC testing using one testing platform, the GeneXpert IV system, and costs may vary for other POC platforms such as Alere-Q, Samba, or Abbott m-PIMA. Nonetheless, we note that the cost of the Cepheid GeneXpert is similar to that of Abbott m-PIMA. Additionally, we only estimate costs of the 4-module GeneXpert instrument; however, it is available in varying sizes (ranging from 4 to 80 modules); this flexibility can enable clinics to reduce the cost per POC test by choosing the instrument size that best fits their testing volume. Future analyses should estimate the costs of implementing varying instrument sizes to identify the optimal machine by clinic testing volume. We estimate the cost of implementing POC VL testing for adults in Kenya, but we conducted time and motion observation on pregnant women. However, we anticipate that the POC testing costs would not vary substantially for non-pregnant women and men. Lastly, this analysis was conducted from a payer perspective and therefore does not account for societal-level costs, including those incurred by patients, to attend clinic visits and undergo viral load testing. We expect that POC testing would result in lower patient costs relative to centralized laboratory testing because patients would not incur transport costs and missed work associated with a second visit to the clinic to receive their VL test results.

The strengths of our analysis include detailed time and motion data based on observations of clinic care, laboratory processes, and adherence counseling. Data collection was carried out at five study sites with varying resource levels, patient volumes, and staff, contributing to the generalizability of our findings. We estimated costs of POC VL implementation for both adults and children living with HIV and we provide estimates of adherence counseling costs separately by viral suppression category.

We estimated POC VL test costs at varying testing volumes and assess the impact of cost-sharing with other diagnostic testing, providing nuanced costs to inform policymakers as POC is rolled out nationally in sub-Saharan Africa. In summary, we find that POC VL technology can be implemented at reasonable costs for clinics with moderate to large patient loads. Future studies are needed to evaluate these aspects of POC testing in hard-to-reach, non-research facilities across various clinic volumes in order for this promising intervention to improve patient health outcomes for people living with HIV.

## Figures and Tables

**Figure 1 diagnostics-11-00140-f001:**
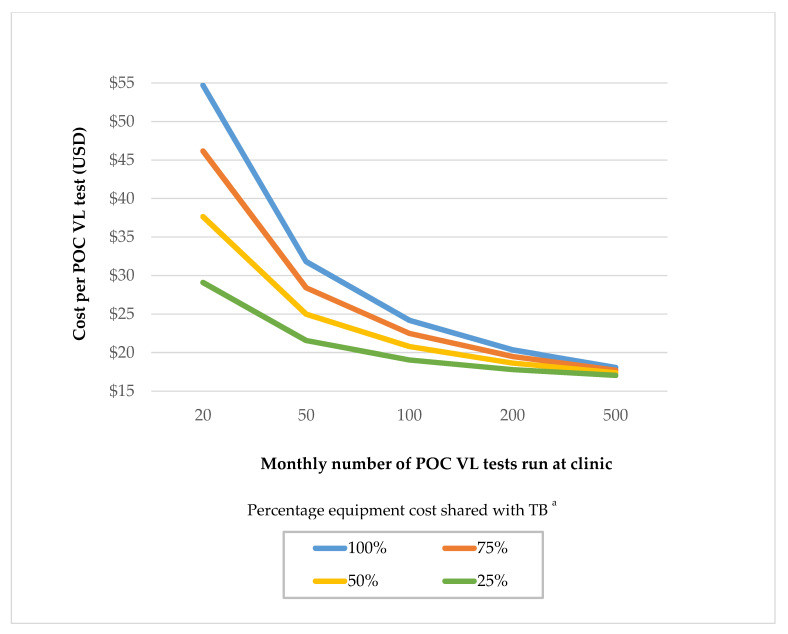
^a^ Sensitivity analysis assessing the potential impact of cost-sharing of POC GeneXpert instruments with TB diagnostics. As POC VL is not yet rolled out, the potential for cost-sharing equipment with other diseases remains uncertain. Therefore, we present costs assuming 25%, 50%, 75%, and 100% of the equipment is used for POC VL testing, with the remaining absorbed by other diagnostics programs like TB.

**Figure 2 diagnostics-11-00140-f002:**
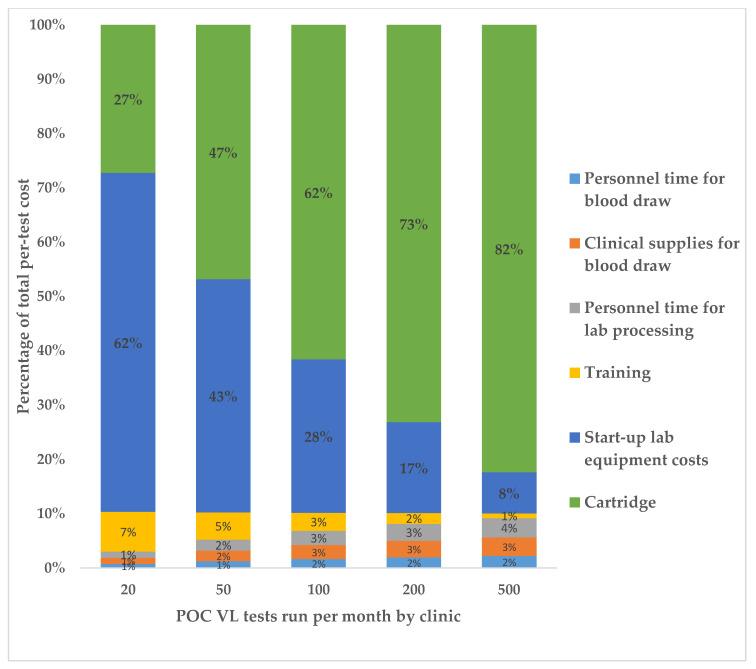
Percentage breakdown of total POC VL per-test costs per category, for varying clinic testing volumes (2020 USD).

**Table 1 diagnostics-11-00140-t001:** Per-test cost of HIV viral load (VL) testing performed as point-of-care (POC) at varying testing volumes and at centralized laboratory facilities in Kenya (2020 USD).

Cost Category	POC Costs by Clinic Testing Volumes (VL Tests Done per Month)					Centralized Laboratory
	20	50	100	200	500	
Personnel time for blood draw ^a^	0.40	0.40	0.40	0.40	0.40	0.40
Clinical supplies for blood draw	0.62	0.62	0.62	0.62	0.62	0.62
Personnel time for lab processing ^b^	0.64	0.64	0.64	0.64	0.64	24.63 [11]
Training	4.00	1.60	0.80	0.40	0.16	
Start-up lab equipment costs	34.15	13.66	6.83	3.41	1.37	
Electricity needed to run machine	0.02	0.02	0.02	0.02	0.02	
Rental space for machine	0.20	0.08	0.04	0.02	0.01	
Cartridge	14.90	14.90	14.90	14.90	14.90	
Total Cost per Viral Load Test ^c^						
POC testing equipment used for VL tests only	54.93	31.92	24.25	20.41	18.12	25.65
75% utilization of POC equipment for VL testing	46.34	28.48	22.53	19.55	17.77	
50% utilization of POC equipment for VL testing	37.75	25.04	20.81	18.69	17.42	
25% utilization of POC equipment for VL testing	29.15	21.60	19.08	17.80	17.07	

^a^ We estimated that personnel time for blood draw took an average of 5 min, at a nurse’s salary cost of $0.08 per minute. For all costs, we assumed 1% wastage of blood samples. ^b^ We estimated personnel time for laboratory processing took an average of 32 min of hands-on work (see Table 4) for a machine that processed 4 samples at once. Therefore, we assumed the personnel time cost for one individual test was 8 min, at a lab technician’s salary cost of $0.08 per minute. ^c^ In each of these scenarios, 1-X% of the cost of the POC equipment is assumed to be absorbed by the tuberculosis (TB) programs. X is either 25%, 50%, 75%, or 100% utilization of POC equipment for VL testing.

**Table 2 diagnostics-11-00140-t002:** Total per-test costs of HIV POC VL testing at varying cartridge costs and clinic testing volumes.

POC VL Cartridge Cost ^a^	Total Cost per POC VL Test, by Clinic Testing Volumes				
	20	50	100	200	500
$12.00	52.03	29.02	21.35	17.51	15.22
$14.90	54.93	31.92	24.25	20.41	18.12
$17.00	57.03	34.02	26.35	22.51	20.22

^a^ Sensitivity analyses of three cartridge costs: $14.90/cartridge as used in the primary analysis (Table 1), which is the current Kenya negotiated cost, $12.00/cartridge to present potential costs with further negotiations, and $17.00/cartridge, which has been cited in previous costing papers in other sub-Saharan Africa settings.

**Table 3 diagnostics-11-00140-t003:** Equipment costs for POC VL testing (2020 USD).

Capital Costs of POC Instruments	Total Cost	Annualized Cost ^a^
WHO negotiated price for one GeneXpert machine with 4-cartridges	26,000.00	5677.00
Calibration install kit (e.g., customs fees, taxes, delivery to site, and calibration)	80.00	17.47
Automated generator	927.00	202.60
UPS GeneXpert back-up system	150.00	32.75
GeneXpert battery system	4000.00	873.36
Total POC instrument costs	31,157.00	6803.18
**Variable Costs**	**Total Cost**	**Unit Cost**
Annual servicing ^b^ (e.g., costs of the servicing kit and the traveling engineer)	6960.00	1392.00
**Fixed Costs (per-Test)**		**Unit Cost**
Single cartridge for POC VL testing (includes reagents/consumables)	Dependent on VL testing volume	14.90

^a^ Annualized unit costs assume a 5-year lifespan at 3% annual discount rate. ^b^ The cost of servicing equipment is dependent on the agreed-upon “servicing package” when purchasing the GeneXpert equipment from Cephied, Kenya. For this analysis, we assumed that servicing would occur on an annual basis.

**Table 4 diagnostics-11-00140-t004:** Time and motion observations on VL testing comparing pregnant women and children and post-test counseling depending on viral suppression status (minutes).

Activity	Personnel Responsible	Average Time per Adult Tested (Range)	Average Time per Child/Adolescent Tested (Range)
Blood draw	Nurse or lab technician (similar salary)	2 (1–3)	10 (6–15)
Complete POC VL test on GeneXpert (4-cartridge)	Lab technician	32 hands-on work, 134 total time	32 hands-on work, 134 total time
Total time		168	181
**Activity**	**Personnel Responsible**	**Counseling for Virally Suppressed Client**	**Counseling for Virally Unsuppressed Client**
Average time for post-test counseling	Adherence counselor, peer educator	12 (5–20)	30 (20–40)
Total cost for post-test counseling ^a^		$0.96 USD	$2.40 USD

^a^ The total cost for post-test counseling was calculated assuming that the cost of adherence counselor time was $0.08 per minute.

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
