# Peer review of "Costs of Point-of-Care Viral Load Testing for Adults and Children Living with HIV in Kenya"

_diagnostics, 2021, doi:10.3390/diagnostics11010140_

Round 1
Reviewer 1 Report
The work will provide readers with valuable information on implementing POC VL testing in resource-limiting countries. Others can mimic the studies and come up with their own estimates. The factors and rationales in calculating the cost were well explained and justified. They have also addressed the limitations of their studies. I wish that they had expanded their work and covered more than one system (Cepheid GeneXpert).
Reviewer 2 Report
Review for “COSTS OF POINT-OF-CARE VIRAL LOAD TESTING FOR ADULTS AND CHILDREN LIVING WITH HIV IN KENYA”
This study aims to estimate the cost of decentralized POC VL testing compared to centralized laboratory testing for adults and children receiving HIV care in Kenya. The researchers have conducted microcosting to estimate the per-patient costs of POC VL testing and carried a time-motion study to assess staff times. The results suggest the high sensitivity of the POC VL per-test cost to the test volume, ranging from $54.71 to $18.09 per test for 20 to 500 VL tests per month, respectively. At a high testing volume, the per test cost of POC VL is comparable to the centralized testing.
General feedback:
The presented methodology for microcosting and the time-motion study is sound. The manuscript can benefit from a detailed proofread and the quality of graphics should be improved. Some simplifying assumptions merits additional discussion in the text and should be incorporated into the limitation section. Table 3 is not referenced in the results section and should be moved to the online supplement. The study can benefit from further sensitivity analysis to different cost components.
Specific questions/suggestions:
- Line 88: What does the “SOC” stand for?
- Line 89: Can you please include the timeline of the clinical trials and the dates of enrollment for children and adults?
- Line 102: Replace “4-80 samples” by “40 to 80 samples”
- Line 103: What’s the reason for popularity of the 4-sample machines in Kenyan clinic? Is this a decision driven by the lower cost of the machine or low test volume?
- Line 110: Small grammatical correction “Costs (2020 USD) were collected from expense reports, staff and expert interviews and the literature, and were divided …”
- Line 119: Can you clarify the month of the “midpoint of the data collection time period (2020)”?
- Line 140: Why were the costs of electricity and space excluded?
- Line 145: Cost-sharing for diagnostic equipment with other diseases can be an effective way to reduce the per test costs, however the final usage will depend on the testing volume for all diseases (driven by the prevalence of each disease). What is the prevalence of TB in areas under study and the estimated TB testing volume at the local clinics in Kenya? Are there other diseases be used for cost sharing?
- Line 166: Why were the cost of “returning results to client” excluded from the study? also does this require the patients to return back to the clinic to receive their results (which could then result in additional transportation costs and loss of wages)?
- Line 169: The role of drug resistant testing is unclear. I assume that the drug resistant test is applicable to both arms of your comparison (both people who receive POC VL testing or go through centralized testing can be eligible to receive drug resistant tests). Is the cost of direct resistant test counted for both arms or only for the centralized scenario?
- Line 184: What was the criteria for reaching information saturation?
- Line 220: Why did the authors make an assumption of 5’ for blood draw personal time ? Wasn't this time estimated via the time motion study?
- Line 222: Again, the word “assumed” here is unclear. If the times are indeed estimated from the time motion study, I suggest clarifying that in the footnote of the table.
- Line 225: 1-X%, X=[0%, 25%, 50%, 75%] ?
- Line 236: The footnote seems redundant. Essential formation should be merged into the figure’s caption.
- Figure 1: The quality of the graphic can be improved. A similar font (and size) should be used in the figure and its legend
- Figure 2: The graphics can go a long way in engaging readers. The quality of this figure can be improved. Also Can you provide parallel figures (in a multi-panel graph) for the cost breakdown under different testing volumes (e.g., 20, 100, 500) per month?
- Table 3: I believe that “annualized” can be replaced with “annual”
- Line 247: Where is the reference for footnote b?
- Table 4: Could you also provide the range of observations? You have included the unit of minutes in the table heading so you don't need to include that in each cell of the table, or can instead remove it from the heading and only included in the corresponding row/cells
- Line 252: Correction: “assuming that the cost of adherence counselor time was $0.08 per minute.”
- Line 191: The results section doesn't include a reference to Table 3. If this information is not presented in the main text, the table can be moved to the supplement.
Additional questions/suggestions:
- Does the implementation of POC VL testing require additional training for clinician and staff to use the GeneXpert machines?
- The patient-incurred costs to attend clinic visits was excluded from the analysis and should be added to the limitations. Can this be different for POC and central testing?
- Excluding details on delays for centralized testing (which may depend on the clinic’s location and its distance from the central lab), the cost of returning the test to the patients (patients transportation cost and loss of wages), and potential loss to follow ups (as a result of delays) in the centralized scenario is a big limitation off this study
- I suggest performing additional sensitivity analysis to activity times, and various cost component.
- Given the projected increase in testing volume over the future years, using large GeneXpert machines to perform more tests maybe applicable. I suggest adding additional sensitivity analysis for the purchase of larger GeneXpert machines and operating them at higher testing volumes.
- Does the cost of small back-up generator and battery for the GeneXpert machine change if larger machines are purchased?
